# Controllable branching of robust response patterns in nonlinear mechanical resonators

Axel M. Eriksson [1] ✉, Oriel Shoshani [2], Daniel López [3], Steven W. Shaw[4,5] & David A. Czaplewski [6] ✉

In lieu of continuous time active feedback control in complex systems, nonlinear dynamics offers a means to generate desired long-term responses using short-time control signals. This type of control has been proposed for use in resonators that exhibit a plethora of complex dynamic behaviors resulting from energy exchange between modes. However, the dynamic response and, ultimately, the ability to control the response of these systems remains poorly understood. Here, we show that a micromechanical resonator can generate diverse, robust dynamical responses that occur on a timescale five orders of magnitude larger than the external harmonic driving and these responses can be selected by inserting small pulses at specific branching points. We develop a theoretical model and experimentally show the ability to control these response patterns. Hence, these mechanical resonators may represent a simple physical platform for the development of springboard concepts for nonlinear, flexible, yet robust dynamics found in other areas of physics, chemistry, and biology.

Systems with robust, yet flexible, dynamics have been proposed as models for a wide variety of complex responses ranging from weather patterns to neural circuits and decision-making[1–7]. Such long-term robust complex response sequences can be described by models from nonlinear dynamical systems in terms of global structures in phase space involving saddle points (or their higher dimensional generalizations) linked by robust dynamics segments[1,8,9]. These constituent responses represent the stable, robust parts of the dynamics that are immune to disturbances, while the saddles allow for flexibility in the selection among the robust parts. In an uncontrolled system, the selection is typically dictated by noise, often resulting in stochastic switching between robust structures. Control of these robust structures necessarily focuses only on the dynamics at the transitions, that is, near the saddles, eliminating the need for active control during the remainder of the response. This type of control strategy, through the use of long-term robust responses, has inspired the emerging fields of physical intelligence[10,11] and soft-robotics[12–14] in which simple actuators generate complex behavior by utilizing the inherent dynamics of their

structural components and material properties[15]. Understanding these systems through accurate models of their responses, would enable unique methods to control them and stimulate further development in these fields.

In this article, we investigate a prototypical nonlinear two-mode micromechanical resonator[16] for which the response exhibits several complex long-term robust dynamic structures (RDSs, described more fully in Section 1 of the Supplemental Information), despite the fact that the resonator is only driven by a single harmonic drive. The RDSs constitute a set of links in phase space, which connect a number of branching points at which the dynamics can switch to another link. The links are dynamic structures resulting from fast-slow dynamics. In this experiment, the RDSs stretch to five orders of magnitude longer than the forcing signal period. Furthermore, the time evolution of the system is attracted to only flow along this network of RDSs, which are robust to noise and small perturbations. Hence, the system evolves along a special kind of "strange attractor," here referred to as robust response patterns (RRPs), constituted by combinations of the RDSs; see

[1]Department of Microtechnology and Nanoscience (MC-2), Chalmers University of Technology, Kemivägen 9, SE-412 96 Göteborg, Sweden. [2]Department of Mechanical Engineering, Ben-Gurion University of the Negev, 84105 Beer-Sheva, Israel. [3]Materials Research Institute, Penn State University, University Park, PA 16802, USA. [4]Department of Mechanical and Civil Engineering, Florida Institute of Technology, Melbourne, FL 32901, USA. [5]Departments of Mechanical Engineering and Physics and Astronomy, Michigan State University, East Lansing, MI 48824, USA. [6]Center for Nanoscale Materials, Argonne National Laboratory, Lemont, IL 60439, USA. ✉e-mail: axel.eriksson@chalmers.se; dczaplewski@anl.gov

Supplementary Information 1.6 for details. Different RDSs can be accessed by adjusting the drive parameters, which results in activating more branching points and generating a plethora of RRPs. Furthermore, we show that it is possible to deterministically navigate between the branching points, via the robust links, by applying short well-timed control pulses close to the branching points. This enables flexible navigation among robust dynamic structures in mechanical resonators.

## Results

### Resonator response

The studied mechanical resonators are $3\,\mu m \times 10\,\mu m \times 500\,\mu m$ mechanically tethered single crystal silicon beams, actuated and detected by capacitive comb drives; see Methods Section for details. The in-plane, flexural mode (Fig. 1a top) is harmonically driven with a near-resonant frequency, $\Omega$, and amplitude, $F$, that is sufficiently large such that the response is in the nonlinear regime and dominated by a hardening Duffing nonlinearity. When the resonator response is driven with sufficiently large amplitude, where the frequency is relatively far detuned from the linear

response, it enables a 3:1 internal resonance (IR) with an out-of-plane torsional mode[16] (Fig. 1a bottom), i.e., the vibrational frequencies satisfy $\omega_{tors} \approx 3\omega_{flex} \approx 3\Omega$ (Fig. 1b). To more clearly visualize the resonator response, we demodulate the resonator output with the driving frequency to obtain the in-phase and out-of-phase quadratures so that all plots and sketches are in the rotating frame of reference (except for the data shown in (Fig. 2c)).

To help understand and fit the theory to the resonator response near the IR, a (simplified) bifurcation diagram is constructed, which maps out the qualitatively different response regions in the $\Omega-F$ drive parameter space (Fig. 1c)[17]. For frequencies outside of the IR, the resonator responds in a large amplitude, stationary state, i.e., a harmonic response with constant amplitude and phase. By tuning the drive frequency from the left (or right) into the IR (Fig. 1c), the system undergoes a saddle-node on an invariant circle (SNIC) bifurcation from the left or a Hopf bifurcation from the right[18], both of which annihilate the high-amplitude stationary harmonic state. As the frequency is driven deeper into the IR, the system undergoes a complex sequence of bifurcations resulting in RRPs that are

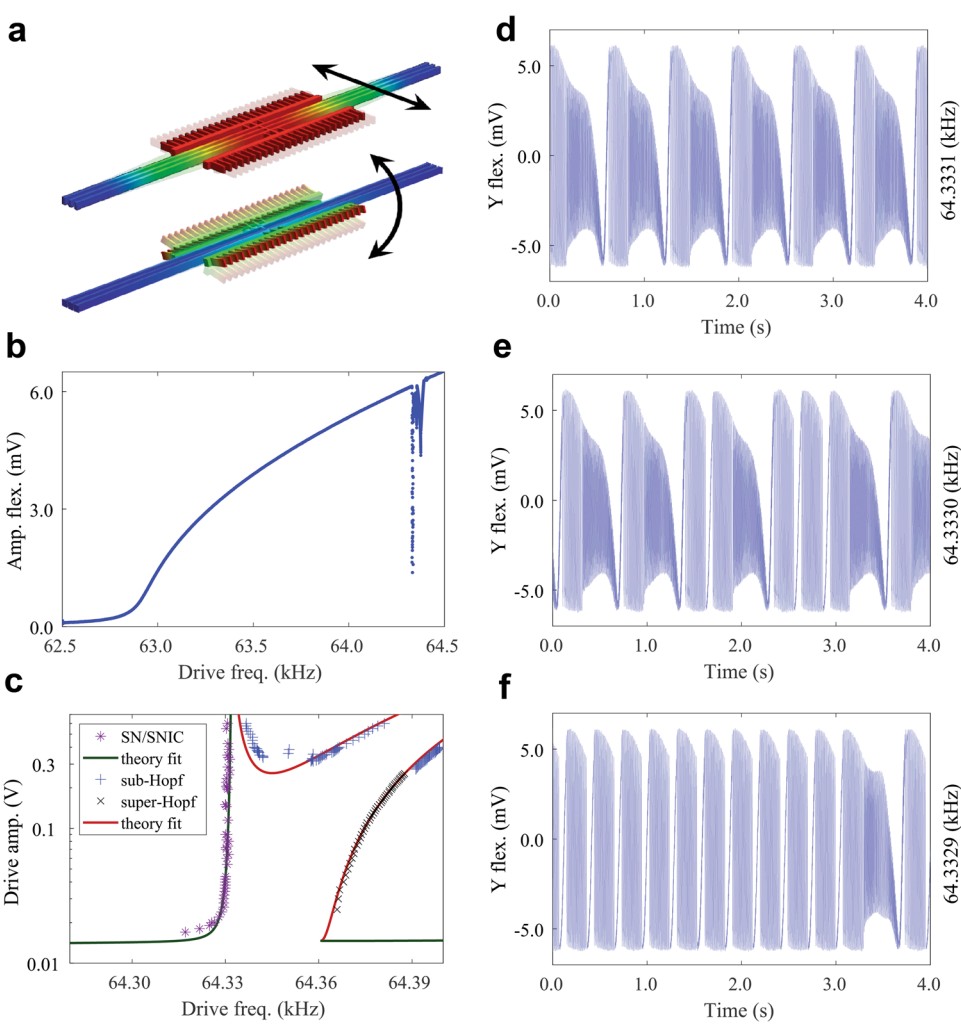

**Fig. 1 | Operating principles and resulting dynamics of the mechanical resonator. a** Simulated in-plane flexural (top) and torsional (bottom) modes of the resonator. Arrows indicate the direction of motion. **b** Resonator (flexural motion) response when driven into the nonlinear regime. Reductions in the average amplitude of the resonator response correspond to a region of internal resonance. **c** The bifurcation diagram predicted by the fitted model, displaying dynamically different regions close to the IR; see SI and ref. [17] for details. The model was fitted to the measured saddle-node (SN), saddle-node on invariant circle (SNIC), subcritical and supercritical Hopf bifurcation points. As the drive frequency approaches the IR

from left (right), the high-amplitude stable state is annihilated by a SNIC (Hopf) bifurcation. As a consequence of subsequent bifurcations, for a sufficiently strong drive, further inside the IR region, the dynamics exhibit long-term RDSs.
**d** Measured resonator response deep inside the IR showing a repetitive execution of an RDS, i.e., a manifestation of an RRP. The drive frequency is indicated on the right of the figure; in this case, 64333.1 Hz. **e** By decreasing the drive frequency slightly (64333.0 Hz), the resonator exhibits a branching mechanism that causes stochastic switching between two distinct RDSs. **f** For yet a slightly lower drive frequency (64332.9 Hz), the resonator is biased to mainly repeat the other RDS.

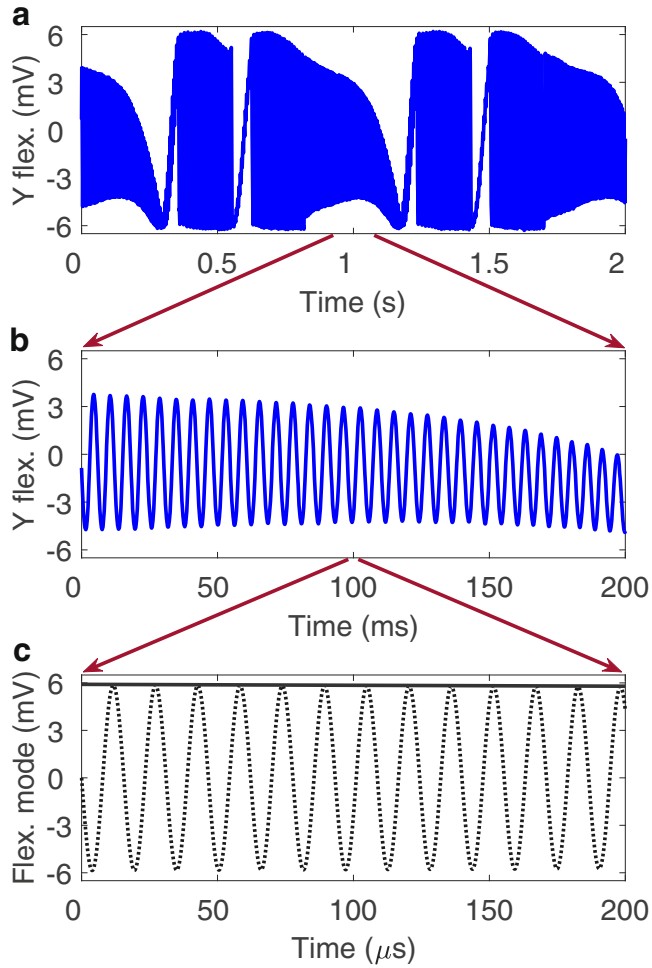

**Fig. 2 | Resonator dynamics on three different timescales of the flexural mode.** Red arrows indicate how the data is displayed at shorter and shorter timescales for subsequent plots. **a** Measured slow evolution of the y amplitude quadrature in the resonator response (i.e., after down-modulation with the drive frequency) showing the repetition of long-term robust responses. The dynamics on this timescale are determined primarily by the dissipation rates of each mode ~1 s. **b** Measured intermediate timescale i.e., zoom-in of (**a**); the modulations of the down-modulated y amplitude quadrature of the resonator response showing dynamics on a timescale ~10 ms determined primarily by the detuning of the flexural mode from its natural frequency. **c** Amplitude $\sqrt{X^2 + Y^2}$ of the flexural mode (solid) and the fast resonator response showing oscillations (dashed) at the timescale of the drive period ~10 μs (i.e., before the signal is down-modulated).

composed of sequences of well-structured long-term response segments, i.e., the RDSs. By varying the drive frequency (and amplitude), the resonator can be biased to execute different, fully repetitive, RRPs in different parameter regions, e.g., (Fig. 1d, f). More interestingly, at the border between these regions, the dynamics display branching mechanisms that cause stochastic switching between the two distinct RDSs, which constitute the RRPs in this region, see Fig. 1e. The branching of these RDSs and how they can be combined into RRPs are the main topics of this article and will be described subsequently.

## Coupled modes
To model the resonator response, we employ a minimalistic generic description of two interacting modes in 3:1 IR, where a single, modal interaction potential term fully captures the dynamics between modes. The tractable model, with the modal coordinate of flexural motion $q_1$ and the modal coordinate of torsional motion $q_2$, is given by:

$$\ddot{q}_1 + \gamma_1 \dot{q}_1 + \omega_1^2 q_1 + \alpha_1 q_1^3 + 3g q_1^2 q_2 = F\cos(\Omega t), \qquad (1)$$

$$\ddot{q}_2 + \gamma_2 \dot{q}_2 + \omega_2^2 q_2 + g q_1^3 = 0, \qquad (2)$$

which is a simplified version of the so-called normal form for this resonance[19,20]. For the present device, the experimentally determined device parameters are: natural frequencies $\omega_1/(2\pi) = 62973$ Hz, $\omega_2/(2\pi) = 192990$ Hz, decay rates $\gamma_1 = 1.885$ Hz, $\gamma_2 = 4.712$ Hz, Duffing nonlinearity of the fundamental mode $\alpha_1 = 2.47 \times 10^{14} \mathrm{V^{-2}sec^{-2}}$, and mode coupling coefficient $g = 2.964 \times 10^{12} \mathrm{V^{-2}sec^{-2}}$, along with the variable drive amplitude $F$ and frequency $\Omega \approx \omega_2/3$; see SI section 2 for further details. Due to the small dissipation rate and the moderate nonlinear coupling strength, the rotating wave approximation can reduce the problem to the slowly evolving dynamics in a 4D autonomous phase space (or 2D for the independent flexural mode when driven far away from the IR). These equations are analyzed and simulated, showing detailed qualitative agreement between the experimental results and the model (see SI section 1.1 and Movie).

To gain intuition into the resonator steady-state response (i.e., long after the effects of initial conditions have passed) using the model, it is important to recognize that the resonator displays dynamics on three well-separated timescales (Fig. 2a–c) (see SI section 1.2). At the timescale $\omega_1^{-1}$, both modes oscillate at the very fast timescale of the drive period ~10 μs (Fig. 2c). To simplify the presentation of the data, these high-frequency oscillations are removed by demodulating the resonator response with the drive signal. The amplitudes and phases of these very fast oscillations vary slowly in time, resulting in amplitude/phase-modulated motions. The amplitude of the flexural mode exhibits modulations on an intermediate timescale of ~10 ms, mainly due to the large detuning of $\Omega$ from its natural frequency, $|\Omega - \omega_1|^{-1}$ (Fig. 2b). The torsional mode amplitude changes much more slowly, on the slow timescale of the dissipation rates ~1 s, $(\gamma_2)^{-1}$ (Fig. 2a). Hence, at the intermediate timescale, the amplitude of the flexural mode exhibits (quasi-) conservative motion, which is influenced by the more slowly varying state of the torsional mode. The frequency of these oscillations at the intermediate scale can be estimated by elliptic integrals (see Supplementary Material section 1.5). Displaying the resonator response on the longest timescale shows the RDSs and the switching between them.

## Branching mechanism
Further analysis (see SI sections 1.3 and 1.4 for details) shows that the branching mechanism that causes the switching between the two RDSs is revealed by treating the amplitude and phase of the torsional mode as slowly varying parameters for the dynamics of the fast flexural mode (Fig. 3a). The mechanism can be understood as follows: Just before a branching event, the trajectory associated with the flexural mode encircles a saddle structure as shown schematically in Fig. 3b. Since there is virtually no dissipation on the intermediate timescale, the flexural mode variables can be thought of as a conservative system in a 2D-phase space, whose morphology is dictated by the slowly varying state of the torsional mode. Hence, the two loops that connect the saddle point to itself separate the corresponding 2D-phase space into three disjoint regions. However, as the torsional mode amplitude slowly drifts, the morphology, and thereby the shapes of the loops, evolve in time, requiring higher dimensions to visualize the full response. As a consequence, an abrupt transition can be induced if a loop deforms in such a manner that it traps the dynamic state; forcing it to instead perform motion inside one of the two loops (Fig. 3d, g). The altered behavior of the flexural mode variables immediately after the transition feed back into the slow torsional variable dynamics and

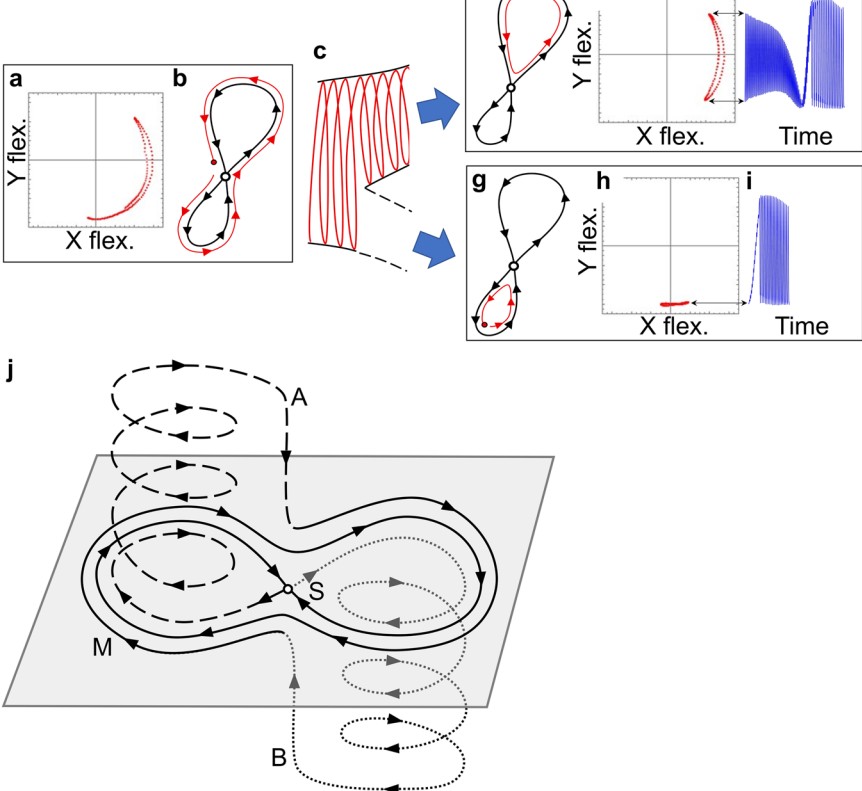

**Fig. 3 | Qualitative description of the branching mechanism.** Immediately before each branching event, the resonator closely encircles the outer part of two loops: **a** measured quadrature of the fast flexural mode, **b** schematic of the corresponding structures in phase space. **c** During the branching event, the fast variables cross (due to the drift of the slow variables) into either the upper (**d**) or lower (**g**) loop, corresponding to the measured responses in (**e**, **h**). The dynamics enclosed by these individual loops will then evolve and feed back into the slow dynamics, thereby executing the corresponding RDSs shown in panels (**f**, **i**). **j** A 3D schematic representation of the branching mechanism where the in-plane oscillatory dynamics (M solid) approaches the saddle structure S, branches either upwards (A dashed) or downwards (B dotted) and eventually gets re-injected into M. Hence, paths A and B represent the two RDSs, which connect the branching point S back to itself (in two different ways). Analogously, the paths depicted in (**f**) and (**i**) correspond to the two paths in 4D space, which link the dynamics of the experimentally observed branching point. The full dynamics of the system is well described as repeated combinations of one or both of these RDSs, e.g., the robust dynamic patterns seen in Fig. 1d–f.

the complete system evolves along an RDS (see SI section 1.7). Even more interestingly, since the state can be trapped by either of the two loops, this mechanism enables the branching of the dynamics into different RDSs, see Fig. 3c. The resonator will perform either response (Fig. 3d–i) until completion and return to the saddle structure. Back at the saddle, the parameters of the system, along with the noise-influenced state trajectory, will dictate the selection of the next RDS to be followed. A schematic of this process can be found in Fig. 3j. Without outside interference, the system will continue to encounter the branching mechanism and the system will continue to execute a single RDS or stochastic combinations of the two (or more) robust structures.

However, as a consequence of this saddle-encirclement mechanism, the resulting RDS that occurs after a branching event can be selected through a small change in the drive amplitude at the branching point, referred to here as a control pulse. With a suitable amplitude control pulse, applied at a critical point in time immediately prior to coming near the branching saddle point, the resonator can be pushed into the desired loop and will therefore execute the selected RDS. To demonstrate experimental control of these robust responses, and thereby confirm the theoretical model of the branching mechanism, the resonator is tuned into a regime where it exhibits stochastic switching between two RDSs. The timing of the control pulses are identified by real-time processing of the critical slowing down of the dynamics, which is a general feature close to a saddle structure, such as

the branching points. When the dynamics only has ≈1 oscillation left before the branching event, either of two control pulses is applied to select the desired RDS (details are provided in SI sections 1.9 and 2). The amplitude of the pulses depends on the details of the branching point, but can easily be found empirically. Note that several solutions to control amplitude and timing can achieve the task of pushing the dynamics into the desired loop. When the control pulses are properly tuned, each of the two pulse types deterministically initiates one of the RDSs that then fully executes, subsequently bringing the system back to the branching point. Similar pulses applied during the robust segments of the dynamics have no appreciable effect on the dynamics. This control is experimentally demonstrated in (Fig. 4a, b) using two different control pulses applied with different time offsets as the dynamics approaches the branching point. Implementing these two control pulses in any selected sequence results in the execution of the attendant sequence of the two RDSs, for instance, the alternating sequence shown in Fig. 4c.

## Discussion

Implementing control strategies for nonlinear systems have been investigated in a wide range of settings. For example, the introduction of small perturbations periodically applied in the OGY method has successfully been used to stabilize and switch between different initially unstable trajectories in chaotic systems. This approach has applications including secure communications[21], gait control of legged

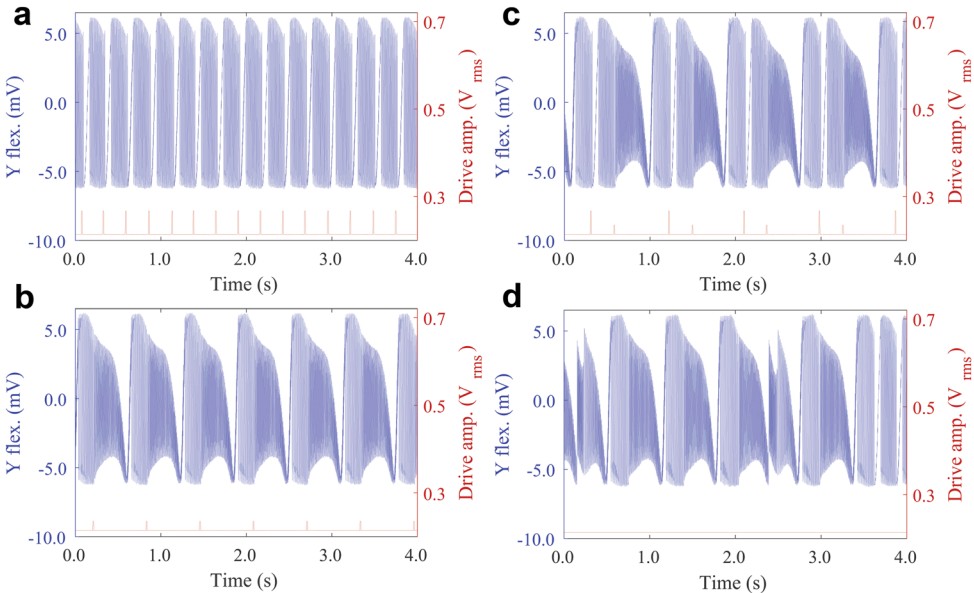

**Fig. 4 | Experimental demonstration of the control pulses at the branching point for the selection of RDSs.** In the switching regime, the system is close to being captured by either of the two loops. The outcome is, therefore, highly sensitive to initial conditions and noise close to the branching point. Hence, by applying a small control pulse (consisting of a change in drive strength of the harmonic drive away from the baseline at 0.214 V and drive frequency fixed at 64333.0 Hz) with a duration of ~10 ms at the branching points (red curve corresponding to the right axes), the dynamics can be steered into either of the two loops (**a** or **b**), which determines the subsequent long-term (~1 s) behaviors. **c** Here, the resonator is forced to alternate between the two RDSs. **d** The phase space exhibits several coexisting dynamic responses, which can be accessed via activation of analogous mechanisms by varying the drive parameters (increasing drive frequency to 64333.55 Hz) without applying control pulses. Note that all RDSs are not necessarily possible to activate in the noise-free model without control pulses.

robots[22], lasers[23], cardiac response[24], and the brain[25]. These methods also include applying small pulses in the vicinity of saddle structures; however, these applications typically involve switching between relatively simple periodic oscillations that occur within a chaotic structure and, of critical importance, do not incorporate long-term robust structures. Additionally, in contrast to the high flexibility of those chaotic systems, where almost any trajectory can be stabilized, the dynamics of the resonator presented here exhibit only a few, but very distinct, robust structures that occur on much longer timescales.

We note that exploration of the surrounding parameter space, either by control pulses or by noise, can lead to the execution of different RDSs, as seen in Fig. 4d. It is crucial to note that activation of the RDSs only occurs close to branching points and cannot be activated at arbitrary points, that is, away from the branching points the response is robust to noise and applied pulses. The presence of several coexisting branching saddles enables different robust responses to be activated and combined into complex RRPs. Interestingly, the new structure revealed in Fig. 4d has not been observed in the noise-free mathematical model (Eqs. (1, 2)). This stresses the importance of combining the understanding gained from models with experimental results of physical systems, since models, while very powerful, can never capture the complete nonlinear dynamics of the latter.

In this regard, our dynamics is more reminiscent of the pulse control scheme proposed in ref. [26] for navigating among heteroclinic sequences in high-dimensional networks of oscillators. Both in the heteroclinic sequences as well as in the RRP presented here, the dynamics is passing from saddle structure to saddle structure. However, in typical heteroclinic sequences, the dynamics is mostly hovering close to the saddles (the local "states")[1] due to critical slowdown, and the links between the saddle states are fast transitions ($t_\text{saddle} \gg t_\text{link}$). On the contrary, the response presented here exhibits comparably fast visits near the saddle structures but long-term oscillatory links (i.e., RDSs) between the saddles ($t_\text{saddle} \ll t_\text{link}$).

Furthermore, the local dynamics of the branching mechanism can be viewed as a more general form of the phenomenon known as resonant capture[27]. In the present system, the dynamics is not restricted to be captured by a loop, but can switch back and forth between the two inner loops and the outer loop (see the movie in the SI). On the other hand, the global structure of the dynamics studied here resembles what is referred to as bursting, where the dynamics alternate between a heavily oscillating state and a calm state. However, in contrast to the flexibility provided by the branching mechanism, the transitions between the states in bursting are caused by bifurcations[28] that lack the possibility of branching the behavior. The inability to drastically affect the response of bursters with small control pulses is also a characteristic of more simple amplitude modulations, for example, those generated by a Hopf bifurcation[29]. Hence, the resonator dynamics presented here is not bursting, but belongs to the more loosely defined family of mixed-mode oscillations (MMOs), distinguishing the present work from previous reports on this topic[30,31].

In this paper, we have presented a micromechanical resonator that exhibits multiple, robust dynamic structures which can be both deterministically and stochastically combined into robust response patterns. This resonator response exhibits three distinct timescales that span five orders of magnitude. A theoretical normal form representation of the micromechanical resonator response accurately predicts these dynamics, and allows one to suggest the magnitude and timing of control pulses needed to control the selection of the robust dynamic structures that compose a given RRP. The experimental results validate the model predictions for controlled switching. We emphasize that the branching mechanism, which we described in this work, only requires the structure qualitatively illustrated in Fig. 3j. Therefore, the same branching mechanism can be observed in various nonlinear systems with similar dynamic structures; examples include NEMS/MEMS[32,33], optomechanics[34], magnomechanics[35,36], and circuit QED[37]. Moreover, in comparison to more complex systems, such as fixed action patterns in animals[38,39], the multifaceted response of the mechanical resonator presented in this work offers a much cleaner and more well-defined repertoire of robust behaviors. We anticipate that this mechanical system and normal form model can offer springboard concepts for understanding and control in more sophisticated systems

such as neural dynamics, soft-robotics, or fixed action patterns observed in animals.

## Methods

The resonator is a microelectromechanical structure (MEMS) consisting of three, 3 μm wide, 10 μm thick, and 500 μm long, doubly clamped, single crystal silicon beams connected at their center to two comb drives. One comb drive is used for forcing and the other for sensing. The MEMS resonator is driven by a sinusoidal signal from a Zurich Instruments UHFLI lock-in amplifier to the forcing comb drive. The main body of the MEMS is DC biased using a lead-acid battery to 6.4 V. A 1 MHz signal output from the lock-in amplifier is added to the DC bias. The MEMS response signal from the sensing comb drive is sent through a transimpedance amplifier (Femto DHPCA-100) and is demodulated and recorded by the same lock-in amplifier. The natural frequencies and dissipation rates of the two modes are determined from open-loop measurements using small driving forces. The Duffing nonlinearity parameter is determined by fitting the amplitude-frequency response to the driving amplitudes, for amplitudes in the nonlinear regime but below those of the IR. The coupling coefficient is determined by fitting the experimental data of the bifurcation points near the internal resonance regime, (Fig. 1c) (for details about the fitting parameters, see ref. [17]). The forcing parameters for the resonator are set to a region of internal resonance where the flexural mode and the torsional mode interact. For a drive voltage of 212 mVrms, the drive frequency is increased through the internal resonance regime with responses seen in Fig. 1d–f. To control the response of the resonator at the branching points, an arbitrary waveform generator function in the lock-in amplifier is used. The output from the arbitrary waveform generator is triggered by amplitude set points of the measured signal coming from the MEMS device. Short voltage pulses are added to the sinusoidal output from the lock-in, i.e. the amplitude of the ≈64 kHz drive signal is slightly increased for a duration of 10 ms. To determine when to apply the stimulus pulses, we measure the period of the fast variable $Y_1$ as the resonator encircles the outer loop, see Fig. 3b. When the resonator approaches the saddle structure, the period of these oscillations becomes longer, which is true for the generic case when the dynamics approach a saddle point or saddle structure of any type, due to the slow behavior nearby[9]. Hence, an indication that the fast variable dynamics is about to hit the saddle structure is therefore that the period of the fast variable oscillations exceeds the experimentally fitted threshold of 9.3 ms (measured each time $Y_1$ passes zero from below). To control the outcome of the branching event, we apply timed pulses. We found that timing was the most critical component of the control pulses. The additional amplitude needed to control the outcomes was estimated in simulations to be 20% of the total amplitude, but the exact values used were fine-tuned experimentally. To execute the upper loop response as shown in Fig. 3d–f, we (i) observe the period exceeding the threshold, (ii) wait 7 ms, (iii) apply a 14 mVrms pulse with a duration of 10 ms. Conversely, in order to execute the lower loop response as shown in Fig. 3g–i, we (i) observe the period exceed the threshold, (ii) wait 2 ms, (iii) apply a 35 mVrms pulse with a duration of 10 ms. The applied pulses change the trajectory of the resonator as it approaches the saddle point and causes the dynamic state to be trapped into one of the two loops, which branches the behavior into different robust RRPs.

## Data availability

The datasets generated during and/or analysed during the current study are available from the corresponding author. The data were fully represented in the figures as plotted lines. The authors do not perform any calculations, summaries, or reductions of the data so that the data could be interpreted differently; therefore, the data will be available upon request.

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

## Acknowledgements
We thank Jeff Moehlis for valuable conversations. D.A.C. performed work at the Center for Nanoscale Materials. Work performed at the Center for Nanoscale Materials, a US Department of Energy Office of Science User Facility, was supported by the US DOE, Office of Basic Energy Sciences, under Contract No. DE-AC02-06CH11357. A.M.E. is grateful for support from Swedish Research Council. S.W.S. is grateful for support from the US NSF grant CMMI-1662619 and BSF grant 2018041. O.S. is grateful for support from the BSF grant 2018041 and the Pearlstone Center of Aeronautical Engineering Studies at BGU.

## Author contributions
A.M.E. developed the branching mechanism concept. D.A.C. and A.M.E. designed the experiment and D.A.C. performed the experiments. A.M.E., D.A.C., D.L., and O.S. analyzed the experimental results. A.M.E., O.S., and S.W.S. analyzed and simulated the theoretical model. D.L. and D.A.C. designed the mechanical resonators used in this study. All authors contributed to the creation of the paper.

## Competing interests
The authors declare no competing interests.
