## [Peer Review File · Nature Communications]

REVIEWER COMMENTS

Reviewer #1 (Remarks to the Author):

This paper investigates complex long-term robust dynamic structures of a nonlinear two-mode micromechanical resonator. The work is interesting and will be a contribution to the field. Therefore, this study is recommended to publish after some corrections and revision:

1. In "II. RESULTS" section, the first paragraph, could the authors add reference after "with an out-of-plane torsional mode"?
2. The authors are suggested to go through the paper carefully to clean up errors, for example, Fig. 2d-f are missing in FIG. 2. in supplement information.
3. In FIG. 1., the varied drive frequency leads to different responses as shown in FIG. 1(d), (e) and (f). Could the authors further explain at what frequency do these three situations occur, and is there a more qualitative or quantitative analysis?
4. In FIG. 2, relations between the frequency and time scales of the signal in the three figures are not clear, especially the region in the two red line with arrows in FIG. 2(b) and (c).
5. The authors claim that this study "offered springboard concepts for understanding and control in more sophisticated systems such as neural dynamics...". Could the authors provide a specific example to illustrate it?

Reviewer #2 (Remarks to the Author):

The paper presents response features spanning timescales in a micromechanical resonator described by coupling between two mechanical modes. The same type of device has been reported previously by the same authors in the context of reports of internal resonance and mechanical frequency combs.

The behavior of the resonator is certainly interesting but not unexpected and such qualitative behavior has been previously observed in <https://doi.org/10.1016/j.physd.2011.01.004> for instance and often not reported in the published literature due to the lack of credible practical applications relegating such phenomena to the domain of scientific curiosity and open-ended exploration or often operational regimes to be avoided in practical devices. While the authors make a brave attempt to link their work to practical applications, the paper does not provide any evidence that this link can be robustly made.

I have no doubt though that the results are ultimately publishable but advise the authors to seek a specialist journal in nonlinear dynamics or MEMS to publish these results; they are not suitable for broad interest journals in my opinion.

Reviewer #3 (Remarks to the Author):

The authors discuss an interesting type of nonlinear dynamics that can occur near internal resonance conditions and discuss ways to externally control the dynamics with relatively short pulses that are generated at well-chosen times. The work thus clarifies the slow dynamic of strongly coupled resonant systems and proposes a novel way to control them. Although I think the work might be accepted for publication, I recommend the manuscript to be improved on the following points before publication:

1. The authors introduce two new terms: robust dynamic structures (RDS) and robust response patterns (RRP). The definition of these terms are unclear, and it is also unclear if there aren't already other names existing for these phenomena (like limit cycles or attractors). I would recommend the authors to reevaluate whether these new terms are really needed and novel, instead of using terms that are common in the nonlinear dynamics community.

If the authors want to keep using these terms, I strongly recommend defining them better/more clearly, preferably with mathematical equations. What conditions does a signal have to be obey to be called a RDS or RRP? What is the difference between an RDS and RRP? Also indicating the RDS and RRP in the figures with labels and arrows would help to clarify the terms.

2. The authors state that the flexural mode exhibits motion at a timescale $|\Omega-\omega_1|^{-1}$. It is unclear to me if it is only the quadrature of the motion of that mode that changes at this timescale or if it is the actual motion of the mode. If you take the quadrature of a mode oscillating at ω_1 w.r.t. a reference frequency Ω then this quadrature will change at a timescale $|\Omega-\omega_1|^{-1}$, even though the actual mode only has frequency ω_1 . So please check carefully if this timescale is not just an artifact of the demodulation procedure and explain this to the reader.

3. The authors state that the slow dynamics occurs at a timescale of the dissipation rate γ_1^{-1} , but doesn't γ_2 also play a role here and isn't the exact timescale more complex?

4. Phenomena similar to RDS/RRP seem to have been earlier observed and analyzed in Iwatsubo, T. et al., J. Sound Vib., Vol. 30 p. 65 (1973) and van der Avoort, C. et al. J. Micromech. Microeng. Vol. 20, p. 105012 (2010). In these works the phenomena are called combination resonances or beating. Although the analysis in the current manuscript is much more thorough, it seems that the RDS/RRP phenomena are very similar (or identical) to these previously reported effects. It is recommended that the authors explain differences and similarities of their work to these prior work in the manuscript.

5. The control strategy is very nice and probably one of the most novel results of the work. It is explained how it was determined when the pulses should be generated using realtime processing, but how was it determined what the amplitude of the pulses should be? Can the authors discuss how to generalize the proposed control strategy?

6. Why is there a DC (or constant RMS) background signal applied in between control the pulses, is that the same as the AC drive signal that is always present?

7. There is little explanation of the experimental settings and device geometry (the reader is referred to other works). Nevertheless, the experimental settings should be described to a level of detail such that they can be reproduced, I am not sure that this is the case (AC and DC voltage levels are needed). Can the authors comment on how crosstalk between the modes in the readout signal is prevented?

8. Can the authors elaborate on whether the type of complex dynamics they describe occurs near any internal resonance (or even away from internal resonance), and on what the conditions are to observe them?

Authors replies to review comments

We thank the reviewers for their constructive comments and suggestions, which have significantly improved our manuscript. We thank reviewers 1 and 3 for their assessment that our work is interesting and will be a significant contribution to the field. We have directly addressed the comments from reviewer 2 regarding previous work. We feel we have fully addressed all comments and look forward to the publication of this manuscript.

We cut and pasted the reviewers' comments into this document. We then included a reply from the authors along with the attendant changes to the manuscript below each comment.

Reviewer #1

This paper investigates complex long-term robust dynamic structures of a nonlinear two-mode micromechanical resonator. The work is interesting and will be a contribution to the field. Therefore, this study is recommended to publish after some corrections and revision:

Reply: The authors thank the reviewer for the analysis of the manuscript, the identification of the interesting nature of this work, and the recommendation to publish this work. We have worked to address all the points pointed out by the reviewer and have made modifications that we hope the reviewer finds satisfactory.

1.

In “II. RESULTS” section, the first paragraph, could the authors add reference after “with an out-of-plane torsional mode”?

Reply: We added the requested reference.

Changes: We added the reference to *Nature communications* 3, 1–6 (2012) after “with an out-of-plane torsional mode”.

2.

The authors are suggested to go through the paper carefully to clean up errors, for example, Fig. 2d-f are missing in FIG. 2. in supplement information.

Reply: We thank the reviewer for pointing out this typo. We have checked the full manuscript and supplemental information and are confident that all references between the manuscript and SI are accurate. This error occurred because we could not couple the figure references between the two overleaf documents (main document and supplemental information).

Changes: We changed the reference in the supplemental from “Fig.2 d-f” to “Fig.1 d-f”.

3.

In FIG. 1., the varied drive frequency leads to different responses as shown in FIG. 1(d), (e) and (f). Could the authors further explain at what frequency do these three situations occur, and is there a more qualitative or quantitative analysis?

Reply: The drive frequencies for Fig. 1 d, e, and f are 64.3329, 64.3330 and 64.3331 kHz, respectively, as indicated to the right of each plot.

Regarding the analysis: A quantitative numerical study is shown in Fig. S5, where red lines are guides to the eye to identify the different regions. Note that these numerical results are for a noise free model (to the level of the numerical accuracy of the simulation). If noise is included in the simulation, the regions will be more vaguely defined.

A qualitative description of where to find the phenomenon is added in Supplementary Material section 1.7.

Changes: We added the frequencies to the figure caption.

Regarding the analysis, we added to the supplemental text near Figure S5: Inherent noise in the system blurs the distinct regions over which a specific structure is executed. We have determined from the numerical and experimental results that the parameter values allow for execution of primarily a single structure to occur even with noise.

4.

In FIG. 2, relations between the frequency and time scales of the signal in the three figures are not clear, especially the region in the two red line with arrows in FIG. 2(b) and (c).

Reply: We have modified the text to clarify which signal is down-modulated or not.

Changes: Fig 2c is updated showing the input signal (dashed line) with a line for the modulated amplitude (solid line).

In Fig 2 caption, we added the highlighted text:

“Resonator dynamics on three different timescales **of the flexural mode. Red arrows indicate how the data is displayed at shorter and shorter timescales for subsequent plots.** a) Measured slow **evolution of the y amplitude quadrature in the** resonator response **(i.e. after down-modulation with the drive frequency)** showing **repetition of** long-term robust responses. The dynamics on this timescale are determined primarily by the dissipation rates of each mode ~ 1 s. b) Measured intermediate timescale **i.e., zoom-in of a); the modulations of the down-modulated y amplitude quadrature of the** resonator response showing dynamics on a timescale ~ 10 ms determined primarily by the detuning of the flexural mode from its natural frequency. c) **Amplitude $\sqrt{X^2+Y^2}$ of the flexural mode (solid) and the fast resonator response showing oscillations (dashed)** at the timescale of the drive period $\sim 10 \mu\text{s}$ (i.e., before the signal is down modulated).”

5. The authors claim that this study “offered springboard concepts for understanding and control in more sophisticated systems such as neural dynamics...”. Could the authors provide a specific example to illustrate it?

Reply:

The authors realize that this sentence in the abstract is not tied to specific examples. Additional or specific examples cannot be added in the abstract because of word limitations. The first paragraph of the introduction to the main article addresses some areas where this work is relevant. Also, at the end of the manuscript, we have addressed some specific examples of how these results may expand understanding of other systems that obey similar dynamics. In particular, variants of the branching mechanism depicted in Fig 3 j, could become a useful concept in the nonlinear-dynamics toolbox to more easily understand and communicate about branching phenomena in dynamic systems.

Changes: Therefore, to improve the organization and reading of the article, we have exchanged the last sentence in the abstract with the last sentence in the conclusion section. The more generic sentence from the conclusion will address the broad Nature Comm

audience, now located in the abstract, and provide some generic ideas of other applications, while the slightly more definitive sentence from the current abstract will be tied to more specific examples in the conclusion.

In the abstract:

“Hence, these mechanical resonators may represent a simple physical platform for the development of springboard concepts for nonlinear, flexible yet robust dynamics found in areas of physics, chemistry and biology.”

Reviewer #2

The paper presents response features spanning timescales in a micromechanical resonator described by coupling between two mechanical modes. The same type of device has been reported previously by the same authors in the context of reports of internal resonance and mechanical frequency combs.

1)

The behavior of the resonator is certainly interesting but not unexpected and such qualitative behavior has been previously observed in <https://doi.org/10.1016/j.physd.2011.01.004> for instance and often not reported in the published literature due to the lack of credible practical applications relegating such phenomena to the domain of scientific curiosity and open-ended exploration or often operational regimes to be avoided in practical devices. While the authors make a brave attempt to link their work to practical applications, the paper does not provide any evidence that this link can be robustly made.

Reply:

To address the uniqueness of the work, we consulted the authors of the review article on MMOs cited in our manuscript: <https://doi.org/10.1137/100791233>

Here is their reply: “MMOs have experimentally been observed in neuronal models and electronic systems, so it is perhaps not that rare in the big world of experimental observations. However, your system (micromechanical resonator) generates them nano-mechanically, and we know of no other mechanical example that generates MMOs. Furthermore, the agreement is very good, indeed, and over a broad range of parameters.”

In light of this, we agree that amplitude modulated response is a common phenomenon in internal resonances, for example, as shown in <https://doi.org/10.1016/j.physd.2011.01.004>. However, the modulated responses in the present work are qualitatively distinct from those in the referenced papers. Specifically, the papers offered by the reviewers show periodic modulation whereas the present paper shows complicated patterns of modulation, effectively chaotic modulation. While chaotic modulation has been observed ([https://doi.org/10.1016/0022-460X\(89\)90682-2](https://doi.org/10.1016/0022-460X(89)90682-2)), the discrete robust structures in the present system have not been reported, to the authors' knowledge. Also, the present work provides a more detailed description of this chaos than any previous work. Furthermore, this understanding allows for implementation of a control scheme, which is also new and is the primary focus of this manuscript. The transition to amplitude modulated response via a Hopf bifurcation is also possible in the system we present in this work, as briefly mentioned in Fig 1c. However, in our view, the novelty in this work is that the internal resonance also exhibits switching between robust long-term structures and not only repetitive amplitude modulated response. Moreover, our device exhibits several robust structures or “links” that can be switched by small control pulses. This type of robust, yet adjustable dynamics is very different from the non-adjustable amplitude modulation presented in <https://doi.org/10.1016/j.physd.2011.01.004>. In this sense, our phenomenon allows for flexible response adaptation via small pulses at the branching points, which is a novel means to alternate between different responses and may be relevant for other systems beyond mechanical systems. Our mechanism could be one possible way for organisms to switch between complex response structures with minimal cognitive input at the branching

points. Because of this, we argue that the presented phenomenon is much more relevant than signals with periodic amplitude modulation.

Changes:

Section III, last paragraph, before the last sentence, we added

“The inability to drastically affect the response of bursters with small control pulses is also a characteristic of more simple amplitude modulations, for example, those generated by a Hopf bifurcation.”

To the same sentence we added a reference to <https://doi.org/10.1016/j.physd.2011.01.004>

Modified the last sentence:

“Hence, the resonator dynamics presented here is not bursting, but belongs to the more loosely defined family of mixed mode oscillations (MMOs), distinguishing the present work from previous reports on this topic.”

2)

I have no doubt though that the results are ultimately publishable but advise the authors to seek a specialist journal in nonlinear dynamics or MEMS to publish these results; they are not suitable for broad interest journals in my opinion.

Reply: We respect the opinion of reviewer #2; however, in our view, the generality and simplicity of our system make the observed branching mechanism potentially relevant for a wide variety of fields ranging from physics to chemistry and biology. Furthermore, we present a simple geometric view of the branching mechanism (Fig. 3j), which allows a broader audience to more easily comprehend the mechanism and add it to their toolbox of nonlinear concepts. We emphasize that the nonlinear structure (Fig. 3j) can be present in generic families of nonlinear systems, and is not tied to the specific equations presented in our paper, although these equations are a simple example that demonstrate the existence of the phenomenon. (The specific model we consider *is* generic for 1:3 resonance and similar MMO dynamics are known to occur in other systems with four states). Furthermore, to our knowledge, such clear MMO dynamics, which we present, have never been experimentally reported in the literature. Although theoretical and numerical investigations constitute a huge body of knowledge, also here, we are not aware of any work which describes the type of switching and controllability which we demonstrate in this work. Therefore, our work's core finding - the possibility to switch between long-term oscillatory structures at key branching points - can be applied to a broad class of dynamical systems and fits well with the broad audience of Nature Communications.

Reviewer #3

The authors discuss an interesting type of nonlinear dynamics that can occur near internal resonance conditions and discuss ways to externally control the dynamics with relatively short pulses that are generated at well-chosen times. The work thus clarifies the slow dynamic of strongly coupled resonant systems and proposes a novel way to control them. Although I think the work might be accepted for publication, I recommend the manuscript to be improved on the following points before publication:

Reply: The authors thank the reviewer for the analysis of the manuscript, the identification of the novelty of this work, and the recommendation to publish this work. We have worked to address all the points pointed out by the reviewer and have made modifications that we hope the reviewer finds satisfactory.

1.

The authors introduce two new terms: robust dynamic structures (RDS) and robust response patterns (RRP). The definition of these terms are unclear, and it is also unclear if there aren't already other names existing for these phenomena (like limit cycles or attractors). I would recommend the authors to reevaluate whether these new terms are really needed and novel, instead of using terms that are common in the nonlinear dynamics community.

If the authors want to keep using these terms, I strongly recommend defining them better/more clearly, preferably with mathematical equations. What conditions does a signal have to obey to be called a RDS or RRP? What is the difference between an RDS and RRP? Also indicating the RDS and RRP in the figures with labels and arrows would help to clarify the terms.

Reply: The resonator response of interest in the present work falls under the definition of being a "strange attractor." However, the reason to introduce the concept of RDSs (which can be used as building blocks to build up RRP) and not to use the term "strange attractor," is that the responses of the device presented here are composed of a set of distinct response structures. This is in contrast to the (more commonly occurring) continuum of oscillatory loops of strange attractors, for example, as in the 'Lorentz system' which has a 'continuum' of unstable periodic orbits. Also, we limit the description of RDSs to be qualitative since a formal definition of RDS with equations is beyond the scope of the present work. We note that many mixed mode oscillations (MMOs) are also composed of sequences of similarly defined structures, namely large and small amplitude oscillations (LAOs and SAOs), that are described only in a qualitative manner, even in the mathematics literature. Please see the added section in SI which aims to better describe these terms and concepts more precisely.

Changes: Introduction, second paragraph, first part is rewritten and now reads:

"In this article, we investigate a prototypical nonlinear two-mode micromechanical resonator for which the response exhibits several complex long-term robust dynamic structures (RDSs, described more fully in Section 1 of the Supplemental Information), despite the fact that the resonator is only driven by a single harmonic drive. The RDSs constitute a set of links in phase space, which connect a number of $\{l\}$ branching points at which the dynamics can switch to another link. The links are dynamic structures resulting from fast-slow dynamics. In

this experiment, the RDSs stretch to five orders of magnitude longer than the forcing signal period. Furthermore, the time evolution of the system is attracted to only flow along this network of RDSs, which are robust to noise and small perturbations. Hence, the system evolves along a special kind of "strange attractor," here referred to as robust response patterns (RRPs), constituted by combinations of the RDSs; see Supplementary Information 1.6 for details. Different RDSs can be accessed by adjusting the drive parameters, which results in activating more branching points and generating a plethora of RRPs. Furthermore, we show that it is possible to deterministically navigate between the branching points, via the robust links, by applying short well-timed control pulses close to the branching points. This enables flexible navigation among robust dynamic structures in mechanical resonators."

At the end of figure caption 3 we added:

" Hence, the paths A and B represent the two RDSs, which connect the branching point S back to itself (in two different ways). Analogously, the paths depicted in (f) and (i) correspond to the two paths in 4D-space, which link the dynamics of the experimentally observed branching point. The full dynamics of the system is well described as repeated combinations of one or both of these RDSs, e.g., the robust dynamic patterns seen in Fig.1 (d)-(f)."

We added a new section to the Supplementary material "Supplementary Section 1.7 Robust dynamic structures and robust response patterns".

2.

The authors state that the flexural mode exhibits motion at a timescale $|\Omega - \omega_1|^{-1}$. It is unclear to me if it is only the quadrature of the motion of that mode that changes at this timescale or if it is the actual motion of the mode. If you take the quadrature of a mode oscillating at ω_1 w.r.t. a reference frequency Ω then this quadrature will change at a timescale $|\Omega - \omega_1|^{-1}$, even though the actual mode only has frequency ω_1 . So please check carefully if this timescale is not just an artifact of the demodulation procedure and explain this to the reader.

Reply: The system is always harmonically driven at frequency Ω and does not evolve freely (undriven), which is why we always down-modulate with the drive signal. To the left and right of the IR region (marked by the bifurcation lines) in Fig. 1 c, this harmonic drive at frequency Ω results in harmonic oscillatory response at the same frequency Ω . In the quadrature frame which oscillates at Ω , the response is thus a stationary point (stationary amplitude). As the dynamics crosses the bifurcations into the amplitude modulated regime, the stationary point becomes unstable. Depending on which type of bifurcation the system crosses, the frequency of the modulated quadratures can often be well-defined (e.g., from a supercritical Hopf or SNIC). For the supercritical Hopf, the dynamics will respond with small periodic oscillations around the previously stable state. If the bifurcation parameter is increased further from the bifurcation point, the amplitude of these oscillations (in quadrature space) will also increase. Deeper inside the IR region (away from these bifurcation borders) in the region of interest for this paper, the quadratures are modulated in a somewhat similar way. In both cases (see figure below), the dynamics is oscillating around an unstable point in the 2D-phase space of the flexural quadrature plane.

Hopf bifurcation

Our paper

Amplitude modulations (red) are not oscillations around the origo in the quadrature plane and can therefore not be removed by a different down-modulation frequency.

Side note: one important difference is that the resulting Hopf dynamics is a stable limit cycle, whereas the oscillation in our work is not a limit cycle, but a dissipative trajectory evolving towards the encircled stable point (under the assumption that the torsional mode would oscillate at the same amplitude forever). However, since the torsional mode is not stationary but drifts on the slow timescale of the dissipation, the flexural mode doesn't have time to reach the stable point before the torsional mode has deformed the landscape of X_1, Y_1 , resulting in the interesting dynamics observed in this article.

Changes: Section II, paragraph 4, sentence 1, we added “steady-state response (i.e., long after the effects of initial conditions have passed)”

Section II, paragraph 4, before the last sentence, we added: “The frequency of the oscillations at the intermediate scale can be estimated by elliptic integrals (see Supplementary Materials).”

Supplementary material: We added a section describing the intermediate frequency.

3.

The authors state that the slow dynamics occurs at a timescale of the dissipation rate γ_1^{-1} , but doesn't γ_2 also play a role here and isn't the exact timescale more complex?

Reply: We do agree that this is a rough, but still quite good estimate of the timescales since the torsional mode effectively is an overdamped oscillator in the rotating frame.

Changes: Main paper, Section II Results, paragraph 4, second to last sentence: We changed “ γ_1^{-1} ” to “ γ_2^{-1} ”

At the end of Supplementary Section 1.2, we added:

“The torsional mode is effectively an overdamped oscillator $\Delta_{IRs} \sim \gamma_{2rs}$ (see Eq. S11-S12) driven by “high frequency” oscillations $\sim |\Omega - \omega_1|$. Hence, the torsional mode drifts towards the steady state dictated by the average value of the flexural mode, which sets a rough timescale of the slow dynamics to $\gamma_2 \sim \gamma_1$.”

4.

Phenomena similar to RDS/RRP seem to have been earlier observed and analyzed in Iwatsubo, T. et al., J. Sound Vib., Vol. 30 p. 65 (1973) and van der Avoort, C. et al. J. Micromech. Microeng. Vol. 20, p. 105012 (2010). In these works the phenomena are called combination resonances or beating. Although the analysis in the current manuscript is much more thorough, it seems that the RDS/RRP phenomena are very similar (or identical) to these previously reported effects. It is recommended that the authors explain differences and similarities of their work to these prior work in the manuscript.

Reply: We have addressed this comment in the response to the first point of reviewer 2. Please refer to our reply there.

Changes: Section III, last paragraph, before the last sentence, we added “The inability to drastically affect the response of bursters with small control pulses is also a characteristic of more simple amplitude modulations, for example, those generated by a Hopf bifurcation.”

To the same sentence we added a reference to <https://doi.org/10.1016/j.physd.2011.01.004>

Modified the last sentence:

“Hence, the resonator dynamics presented here is not bursting, but belongs to the more loosely defined family of mixed mode oscillations (MMOs), distinguishing the present work from previous reports on this topic”

5.

The control strategy is very nice and probably one of the most novel results of the work. It is explained how it was determined when the pulses should be generated using realtime processing, but how was it determined what the amplitude of the pulses should be? Can the authors discuss how to generalize the proposed control strategy?

Reply: The theory provided the timing for the pulses, while their amplitudes were determined empirically. The most critical aspect, as discussed in other areas, is the timing of the pulse as determined from the theory. A general guide for determining the required amplitude would be to start with a small pulse and increase the amplitude until the desired effect is observed. As with many experiments, a precise value is difficult to estimate from a noiseless model derived from the theory.

Changes: Changed Section II Results, from:

“By real time processing of the dynamics (details are provided in the SI sections 1.6 and 2), the branching point is detected as the system approaches it and one of two short control pulses is applied at the appropriate time. ”

to: “The timing of the control pulses are identified by real time processing of the critical slowing down of the dynamics, which is a general feature close to a saddle structure, such as the branching points. When the dynamics only has ≈ 1 oscillation left before the branching event, either of two control pulses is applied to select the desired RDS (details are provided in the SI sections 1.9 and 2). The amplitude of the pulses depends on the details of the branching point, but can easily be found empirically. Note that several solutions to

control amplitude and timing can achieve the task of pushing the dynamics into the desired loop. When the control pulses are properly tuned, each of the two pulse types deterministically initiates one of the RDSs that then fully executes, subsequently bringing the system back to the branching point. ...”

Added to section 2 of the supplemental:

“We found that the timing was the most critical component of the control pulses. The additional amplitude needed to control the outcomes was estimated in simulations to be 20% of the total amplitude, but the exact values used were fine tuned experimentally.”

6.

Why is there a DC (or constant RMS) background signal applied in between control the pulses, is that the same as the AC drive signal that is always present?

Reply: The DC background signal that you reference between the control pulses is the amplitude of the AC drive signal that is always on. The pulses are changes in the amplitude of the AC signal applied to the resonator.

Changes:

In Fig4 caption, we changed

“Hence, by applying a small control pulse of duration ~ 10 ms”

To

“Hence, by applying a small control pulse (consisting of a change in drive strength of the harmonic drive away from the baseline at 0.214 V and drive frequency fixed at 64333 Hz) with with duration ~ 10 ms”

For Fig4 caption d) we added

“ (increasing drive frequency to 64333.55 Hz)”

The red line in Fig4d has been updated (previous value was showing peak-to-peak and not RMS.)

7.

There is little explanation of the experimental settings and device geometry (the reader is referred to other works). Nevertheless, the experimental settings should be described to a level of detail such that they can be reproduced, I am not sure that this is the case (AC and DC voltage levels are needed). Can the authors comment on how crosstalk between the modes in the readout signal is prevented?

Reply: We thank the reviewer for highlighting that more details are needed here. We have extended the SI section 2 accordingly.

Regarding cross talk concerns, we find that the single modal interaction term captures the full dynamics of the interplay between the two modes as demonstrated through the model and control of the system. In terms of the experimental readout of the two modes, they have well separated frequencies that allow for sufficient discrimination.

Changes: We added additional dimensions of the MEMS device in the supplement and referenced the schematic in Fig 1a in the main text. We also added parameter values used to drive the MEMS to supplemental section 2.

In the main text, added to the results section second paragraph:

“..., where a single, modal interaction term fully captures the dynamics between modes”

8.

Can the authors elaborate on whether the type of complex dynamics they describe occurs near any internal resonance (or even away from internal resonance), and on what the conditions are to observe them?

Reply: Since the model is the normal form for a 3:1 resonance, the phenomenon is generically possible close to a 3:1 internal resonance condition in systems with very light damping in both modes. There are, of course, some conditions on the drive amplitude and frequency that depend on the mode frequency detuning and other parameters. Such conditions may well exist for other internal resonances. Also, the branching mechanism itself could in principle take place in other situations in other systems as long as a number of conditions are met, that are more fully described in the added material.

Changes: We added a whole new section to the Supplementary material “Supplementary Section 1.7.

REVIEWERS' COMMENTS

Reviewer #1 (Remarks to the Author):

I have no further questions, I suggest to accept it.

Reviewer #2 (Remarks to the Author):

The authors have responded to the queries and I agree that the results represent a publishable advance of interest to the peer community in MEMS/NEMS and nonlinear dynamics. There is also novelty in the control scheme shown. The discussion of observations of nonlinear dynamics on multiple timescales in MEMS/NEMS including similar systems is well established though.

I remain unconvinced on the overall impact of the work and its practical relevance and therefore cannot recommend acceptance in Nature Communications.

Reviewer #3 (Remarks to the Author):

The authors have addressed my questions well, I recommend publication of this interesting work.

Authors replies to review comments

We thank the reviewers for their constructive comments and suggestions, which have significantly improved our manuscript. We thank reviewers 1 and 3 for their assessment that our work is ready for publication. We have directly addressed the comments from reviewer 2.

We cut and pasted the reviewers' comments into this document. We then included a reply from the authors along with the attendant changes to the manuscript below each comment.

Reviewer #2

The authors have responded to the queries and I agree that the results represent a publishable advance of interest to the peer community in MEMS/NEMS and nonlinear dynamics. There is also novelty in the control scheme shown. The discussion of observations of nonlinear dynamics on multiple timescales in MEMS/NEMS including similar systems is well established though.

I remain unconvinced on the overall impact of the work and its practical relevance and therefore cannot recommend acceptance in Nature Communications.

Reply: We have modified the conclusions to include examples of systems that display the bifurcations observed in our manuscript that are relevant to current research in a wide range of fields.

“We emphasize that the branching mechanism, which we described in this work, only requires the structure qualitatively illustrated in Fig. 3(j). Therefore, the same branching mechanism can be observed in various nonlinear systems with similar dynamic structures; examples include NEMS/MEMS, optomechanics, magnomechanics, and circuit QED.”

These applications have several references associated with them in the manuscript, but are not reproduced here.